# Looking for In Vitro Models for Retinal Diseases

**DOI:** 10.3390/ijms221910334

**Published:** 2021-09-25

**Authors:** Margherita Alfonsetti, Vanessa Castelli, Michele d’Angelo, Elisabetta Benedetti, Marcello Allegretti, Barbara Barboni, Annamaria Cimini

**Affiliations:** 1Department of Life, Health and Environmental Sciences, University of L’Aquila, 67100 L’Aquila, Italy; margherita.alfonsetti@guest.univaq.it (M.A.); vanessa.castelli@univaq.it (V.C.); michele.dangelo@univaq.it (M.d.); elisabetta.benedetti@univaq.it (E.B.); 2Dompè Farmaceutici Spa, Via Campo di Pile, 67100 L’Aquila, Italy; marcello.allegretti@dompe.com; 3Faculty of Biosciences and Technology for Food, Agriculture and Environment, University of Teramo, 64100 Teramo, Italy; 4Department of Biology, Sbarro Institute for Cancer Research and Molecular Medicine, Temple University, Philadelphia, PA 19122, USA

**Keywords:** cellular models, in vitro, retinal degeneration, 3D models, organoids

## Abstract

Retina is a layered structure of the eye, composed of different cellular components working together to produce a complex visual output. Because of its important role in visual function, retinal pathologies commonly represent the main causes of visual injury and blindness in the industrialized world. It is important to develop in vitro models of retinal diseases to use them in first screenings before translating in in vivo experiments and clinics. For this reason, it is important to develop bidimensional (2D) models that are more suitable for drug screening and toxicological studies and tridimensional (3D) models, which can replicate physiological conditions, for investigating pathological mechanisms leading to visual loss. This review provides an overview of the most common retinal diseases, relating to in vivo models, with a specific focus on alternative 2D and 3D in vitro models that can replicate the different cellular and matrix components of retinal layers, as well as injury insults that induce retinal disease and loss of the visual function.

## 1. Introduction

The retina comprises numerous light-sensitive cells (rods and cones) and other nerve cells that receive and organize the visual information. The eye is an organ that can create a focused two-dimensional image of the world on the retina, which can transform that image into electrical neuronal impulses to the brain to create visual insight [1]. Because of its crucial role in vision, retinal pathologies are the main causes of visual alteration and blindness in the world [2]. Currently, great efforts are being taken to understand mechanisms underlying retinal pathologies and to test the effects of new treatments by mainly using animal experimental models [3]. However, due to ethical concerns, the newest in vitro models of the retinal disease began to replace the in vivo studies or, in any case, contributed toward reducing the use of animals in this field of research [4]. This review collected in vivo models of the most common retinal diseases and alternative 2D and 3D in vitro models created to replicate physiological and pathological conditions, taking advantage of the increased knowledge on retinal anatomy, its cell components, and extracellular matrix composition.

### Retinal Cells and Function

The retina is a multi-layered structure of the human eye formed by different neuronal cell classes associated with Müller glial cells [5] (Figure 1). The choroid is the vascular layer of the eye, containing connective tissues and vasculature, lying between the retina and the sclera, and it is essential to deliver oxygen and nourishment to the outer structures. The choroid can be divided into four layers: the Haller’s layer, which is composed of bigger diameter vessels, the Sattler’s layer, formed by smaller vessels, and the capillaries’ layer, named choriocapillaris [6]. Bruch’s membrane is the extracellular matrix interposed between the choroid and retinal pigment epithelium (RPE) and it is crucial for their structures and functions [7]. The following layer is constituted by cells of the RPE. This epithelium is one cell layer in thickness, and it is crucial for visual functions, for preserving the structure and physiological activities of near tissues and its integrity is fundamental for photoreceptor activity [8]. RPE cells participate in the phagocytosis of photoreceptors outer segments, are involved in the directional transport of nutrients and removal of waste products from photoreceptor cells, and visual pigment transport and regeneration. Alterations of RPE due to age or chronic dysfunctions are associated with retinal degeneration pathologies, such as age-related macular degeneration (AMD) and retinitis pigmentosa (RP) [9,10]. RPE cells extend apical microvilli (maintained by an actin core) to enclose the external segments of photoreceptors with a major length of the outer segment covered by the microvilli: larger for rods than cones [11,12]. For cell–cell adhesion and communication, the apical junctional complexes (constituted by tight, adherent, and gap junctions) at the level of lateral membranes of RPE cells play a crucial role. Furthermore, this barrier function is also paracellular because it also prevents the passage of ions and molecules, making RPE an important portion of the outer blood–retinal barrier (BRB) [13,14]. The reason why this epithelium is defined as “pigmented” is that the melanosome is a prominent organelle located apically in the cell. Melanosome is a body that synthesizes and stores melanin that contains structural proteins, enzymes, and ion channels useful for melanin synthesis. RPE melanin’s function is to absorb light that has passed across the photoreceptors’ retinal layer, defending against reflected light that would then damage the final image. For this reason, melanosomes appear apically in the cell and this compartmentalization is reduced with age where melanosomes bind lipofuscin granules. Moreover, melanin shows an antioxidant function against reactive oxygen species, participating in iron homeostasis [15]. The visual information arises from the conversion of light to membrane potential by photoreceptors [16]. Before being absorbed by photoreceptors, light passes all the retinal layers. At this point, the signal is transduced to another kind of glutamatergic neurons named bipolar cells at the level of the outer plexiform layer, and this connection is modulated by horizontal cells [17,18]. Bipolar cells form synapses, in return, with amacrine cells, and are connected with ganglion cells that transmit the information to the brain through their prolonged axons that form the optic nerve [19]. Rods and cones are two types of photoreceptors in the human retina. Rods primarily contribute to night-time vision (scotopic conditions), whereas cones predominantly participate in day-time vision (photopic conditions), but the mechanisms that support phototransduction are similar [20]. In the human eye, three different types of cone cells exist, distinguished by their response pattern to the light of different wavelengths [21]. Bipolar cells can be distinguished into two classes: rod and cone bipolar cells. Cone bipolar cells generally contact cone photoreceptors, whereas rod bipolar cells mainly synapse with rod photoreceptors [22]. Retinal homeostasis is maintained by the activity of support cells that include Müller cells and astrocytes. Glial cells raise and recycle γ-Aminobutyric acid and glutamate, take up potassium from the extracellular space, and filter metabolites among blood vessels and neurons. Moreover, these cells are involved in the maintenance of BRB. Retinal dysfunctions and subsequent visual loss are mostly associated with the lack of activity and death of the cells described above, contributing to the retinal function and support [23].

## 2. Retinal Pathologies and In Vivo Models

Retinal pathologies such as diabetic retinopathy (DR), RP, AMD, and retinoblastoma (RB) represent some of the most common causes of visual impairment and blindness. The loss of the photoreceptor cells is evident from optical coherence tomography scans, and from histological methods in in vivo models [24,25]. The alteration can be detected by observing the formation of rosette-like structures peculiar in RB, DR, and RP. Neuronal degeneration in the macula is one of the hallmarks of AMD leading to the loss of central vision [26]. On the other hand, photoreceptor degeneration in RP determines primarily the loss of peripheral vision [27]. Another common eye pathology is glaucoma, where the optic nerve appears damaged. This is caused by fluid accumulation that determines an increase in eye pressure [28]. The next paragraphs will describe the most common eye diseases and relative animal models that can replicate pathological conditions.

### 2.1. Age-Related Macular Degeneration and Diabetic Retinopathy

The ocular vascular system is an effective system able to encounter retinal metabolic necessities. The photoreceptor layer acquires oxygen and nutrients by the diffusion, from the choriocapillaris through the RPE because it is avascular [29]. For this reason, much evidence has been shown that pathologies target the eye’s blood circulation [30].

For example, a microangiopathy induced by hyperglycemia is involved in DR. The pathogenesis is a degeneration of pericytes, the proliferation of endothelial cells with a subsequent thickening of the basal membrane, and vessels’ occlusion [31]. Together with an alteration in blood cells, the consequence is an increased hypoxic injury in the retina, causing a spread in the release of angiogenic factors including the vascular endothelial factor (VEGF) that is the main cause of damage in the BRB [32].

In vivo models of DR include animals pancreatectomized or treated with alloxan [33] or streptozotocin (STZ) [34], high-galactose diets, laser, or chemical damage to the eye. The most used methodology is STZ administration because it determines the fastest disease development. Alloxan is less efficient in generating diabetes, and dietary methods require longer time interval for disease progression. Moreover, there are five known genetically manipulated DR mouse models: Ins2^Akita^, nonobese diabetic (NOD), db/db (Lepr^db^), Kimba, and Akimba. Due to the complexity of DR pathogenesis and progression, researchers suggest that the combination of genetic and induced models may mimic better DR features. An example is the Akimba mouse that is obtained from Kimba mice, which overexpress VEGF, and the Akita mice, a spontaneous type I diabetes model. Then, a model can be generated that has many traits of both early and late phases of the DR. To date, the available models mostly represent non-proliferative DR characteristics, such as microaneurysm and retinal degeneration, and no neovascularization, the main feature of proliferative DR [35].

Another pathology that involves not only the retina, but also the interconnected tissues (such as the choroid and the RPE) is the AMD [36]. In particular, this disease is a focal degeneration affecting the central part of the retina called the “macula”. The early stages of AMD include clinical signs, such as drusen and abnormalities of the RPE. Late-stage AMD can evolve in the neovascular form also called “wet” and non-neovascular form known also as “dry”. Early AMD is commonly asymptomatic, but when the pathology evolves, the central vision is affected, and the progression is rapid for the neovascular form and slower for the “dry” form. The wet form is characterized by pathological new vessels arising mostly from the choroid with the leakage of fluid into and under the retina and the RPE, RPE detachment, the presence of hard exudate or subretinal fibrous scar tissue. A risk factor for the development and progression of the “dry” form of AMD and the subsequent progressive atrophy of the RPE and overlying retina is the accumulation of lipofuscin-containing deposits called “drusen”. This atrophy can progress as multifocal or unifocal and can surround but spare the central macula [37].

In vivo models of AMD can be obtained in mice, rats, rabbits, pigs, and non-human primates. The advantage of using rodents is the low cost, quick disease progression, and the ability to perform genetic manipulation. However, one distinct disadvantage of mice and rats is the lack of an anatomical macula. Using mouse models, researchers were able to recreate many of the AMD histological features such as the thickening of Bruch’s membrane, the accumulation of subretinal deposits as well as inflammatory cascade induction together with macrophages and microglia activation [38,39]. These features were obtained using gene therapies, but also laser-induction techniques and high-fat diets [40]. For example, mice can be genetically manipulated for the complement factor pathway (complement factor H, C3a, and C5a knockout or overexpression of C3) the expression of chemokines (knockout for Ccl2, Cxcr2, and Cxcr1), or intrinsic antioxidant mechanisms (Sod1 and Sod2 activity and expression) [41].

### 2.2. Glaucoma

Glaucoma is the major cause of bilateral blindness. The pathogenesis of this disease is not clear, but the main reasons for RGCs degeneration are the elevated levels of intraocular pressure (IOP). Notably, the weakest region of the eye under pressure is the lamina. The sclera results perforated at the level of the optic nerve, whereas RGCs axons leave the eye [28]. This leads to an alteration of the axonal transport that blocks the retrograde supply of important trophic factors to the RGCs from the related neurons of the lateral geniculate nucleus [42].

Most of the therapeutic approaches for glaucoma involve drugs and surgical approaches to reduce the IOP. For this reason, the majority of animal models of glaucoma present optic nerve damage induced by ocular hypertension [43].These include large animals as well as small animals, such as rodents. The disease can be spontaneous, chemically or genetically determined, or surgically induced with lasers or episcleral vein cauterization. Animal models genetically modified are advantageous for experiments focused on the effects of elevated IOP over long periods without the use of experimental procedures that can lead to inflammation [44]. Ocular hypertension is proper of some rodent strains, for example, the TDBA/2J mouse strain show eye damage by the age of 9 months, characterized by the death of the RGCs, optic nerve atrophy, and visual defects. Transgenic models can express mutated Myoc and modified proteins implicated in the pathogenesis of glaucoma (optineurin and Cyp1b1) [45]. To induce a stronger pathology, a different form of stress such as diet modifications and the addition of further genetic manipulations can be introduced [46]. There is no single experimental model that could be suitable to cover the whole pathogenetic aspect of glaucoma, but these models can be used to separately observe the different aspects of the pathology and the effect of treatments [47,48].

### 2.3. Retinitis Pigmentosa

RP is a genetically determined degeneration that affects the retina bilaterally and the degeneration occurs mostly in the photoreceptor layer. The progressive reduction of retinal activity leading to retinal atrophy is determined by the increase in apoptotic events in the retinal layers [49]. Rods are the first cells affected with the subsequent degeneration of cones. Vision loss starts with the development of night blindness and the lack of the visual field [42,43]. These clinical manifestations depend on the gravity of photoreceptors dysfunction, and the loss of central vision appears at the end of the disease course [50]. Histologically, at the retinal level, there are many signs of optic atrophy: pallor of the optic nerve, constricted vessels, and the migration of pigment from the RPE cells disintegration into the interstitial spaces (bone-spicule pigmentation). The visual field alteration begins from the periphery before affecting the center of the eye during the disease [51]. Nowadays, scientists have shown at least 80 loci where mutations were associated with the conditions mentioned before. In particular, the rhodopsin gene that causes the 25% of dominant RP, the USH2A gene that has been associated with the 20% of the recessive disease (Usher’s syndrome type II included), and the RPGR gene that includes the 70% of X-linked RP [52,53]. RP models include mice, rats, rabbits, pigs, zebrafish, and nonhuman primates. Mice and rats allow performing low-cost experiments in a short period and permits genetic manipulations. Nowadays the rd mouse is the most used model for recessive RP. In this model, photoreceptor cell death is both early and rapid. Another model is the VPP mice that contain a three-point mutation (P23H, V20G, and P27L) [54]. These animals develop photoreceptors with the outer segments that never reach a normal length together with photoreceptors aberrant number. Genetic manipulation in mice to mimic the pathophysiology of RP involved other genes, for example, rd4, rd8, rd10, and RPE65 [55,56]. Animal models have led to a better understanding of the pathological mechanisms and participated in the development of novel therapies (in particular gene therapies) for this genetic disorder [57].

### 2.4. Retinoblastoma

Tumors involving the retina can be distinguished between those directly affecting the retina structure and those targeting choroid and the RPE. Alteration of retinal cells into malignant phenotype appears under the age of 3 years. Retinoblastoma (RB) is the most common primary intraocular malignancy of childhood; it is most the time bilaterally and it is frequently multifocal. RB is usually located inside the retina, but the tumor can be endophytic, when it is situated into the vitreous cavity, or exophytic, in case it appears in the subretinal space [58].

There are two forms of RB, genetic and non-genetic. The genetic form is caused by the inactivation of the two alleles of the retinoblastoma (Rb gene, and this leads to the imperfect translation of the retinoblastoma protein (pRB). pRB is the main onco-suppressor gene that plays a role in the cell cycle progression, DNA replication, and terminal differentiation [58]. Activity loss of this pRB in retinal progenitor cells induces the alteration of the cell cycle leading to aberrant cell proliferation [59].

During the last 30 years, numerous genetic murine models of RB have been studied and differed for a lower or higher similarity with the human form of the tumor. RB gene knockout together with the loss of p107, p130, p53, using promoters of Chx10, Pax6, and Nestin, showed morphological changes similar to the human retinoblastoma conditions [60,61]. In fact, it was shown that in post-natal murine retinas and their explant cultures knocked out for RB, p107 gene was upregulated, unlike the human retina, which supported the hypothesis that p107 compensates for the lack of RB [62]. Moreover, the lethality of RB knock-out during the development leads to the introduction of the Cre-Lox technology in the retina [60]. The development of the RB xenograft model allowed to overcome the obstacle of time delay in the pathology development and the appearance of symptoms that is the main weakness of genetic models. Furthermore, xenograft models provide a faster in vivo evaluation of the cell lines and tumor tissue’s tumorigenic potential in immunocompromised animals. It was investigated that transplanted Y79 cells showed metastatic characteristics invading the retina, the subretinal space, the choroid, the optic nerve head, and the anterior chamber of the eye progressing in the brain [63]. On the contrary, WERI-RB cells determine localized tumors with only at later stages an anterior choroidal invasion [64]. The cells injection can be intravitreal, subretinal, and intracameral with a cell number ranging between 10^3^ and 10^7^. Xenograft models are widely used to evaluate the efficacy of new chemotherapeutics ad photodynamic therapies [65].

## 3. From Animal Studies to In Vitro Models

Investigations using animal models of human retinal disorders determine many recognizable benefits. In fact, humans, and animals in general, are very complex organisms and organs that own distinct physiological functions. Connections between organs involve a wide range of hormones, circulating factors, cells, and continuous crosstalk between cells in every tissue. In vivo models of disease (in particular, mammals) have remarkable anatomical and physiological similarities with humans, thus are useful to investigate a wide range of mechanisms and assess novel therapies for human retinal diseases before the application in humans [66]. Exhaustive molecular, histopathological, and electrophysiological investigations that cannot be performed in humans, can be executed in in vivo models contributing to understanding disease mechanisms that occur in retinal disease [67]. The animal models can be used to perform drug treatments or gene therapy to counteract photoreceptor degeneration [68]. Using animals, researchers can better study the safety, toxicity, and efficacy of a drug candidate mimicking their framework complexity [69]. Nevertheless, the use of animals leads to a remarkable improvement in our comprehension of the onset of retinal diseases; yet, numerous features do not adequately reflect the conditions observed in humans. For example, the macula is not present in the mouse retina, and this leads to the impossibility of using mice as macular degeneration in vivo models [70]. Furthermore, the eye of the mouse is different from the human eye for its dimensions, but also for what concerns the rod-to-cone ratio. Notably, the photoreceptor to RPE cell ratio is around seven-fold lower in the peripheral and central areas in the human retina in comparison with the same regions in the mouse retina. Moreover, in humans, only 3% of human RPE cells are binucleate, differently from mouse RPE cells that appear 35% binucleate. Furthermore, the Bruch’s membrane thickness differs between mice and humans’ eyes by about three- to six-fold and in the mice, the membrane is thicker in the periphery in comparison with the central area [71]. In addition, in vivo studies require time and many resources, advanced personnel training in animal handling, and maintenance fees. Then, we have to also consider the ethical aspects, in fact, scientists attempt to perform in vivo studies only when there are no alternatives or when they already achieved preliminary studies, in order to limit the use of animals. The in vitro approach is the most commonly used by pharmaceutical industries for large-scale investigations because of the ease of culture, besides the fact that there is no need to submit animal protocols to the competent authorities as well as economic considerations [72]. Therefore, the next paragraphs will focus on 2D and 3D in vitro models.

## 4. Alternative 2D In Vitro Models

Two-dimensional (2D) models, since the 1900s, have represented the means used to culture cells, playing a crucial role in research, despite many limitations. Cells can only be expanded in two dimensions; cell shapes are flat and elongated and they are not able to represent tissues in vitro [73]. Nevertheless, 2D cultures are still used in research because of the high reproducibility of results, ease of use, low cost, and applicability for long-term experiments and large-scale studies [74].

### 4.1. Neoangiogenesis In Vitro Models

Pathological neoangiogenesis involves the growth of blood vessels from preexisting vessels. This mechanism is regulated by numerous modulators: pro- and anti-angiogenic factors [75]. Neoangiogenesis represents the main cause of vision loss in proliferative retinal diseases (proliferative DR and AMD) [76,77]. In these pathologies, an increased vascular permeability and choroidal neovascularization, controlled mainly by the VEGF that is upregulated under hypoxia conditions, and produced by RPE, is observed. A large number of growth factors, such as the basic fibroblast growth factor (bFGF), angiopoietins, PE-derived factor (PEDF), and adhesion molecules also play a pivotal role in these processes [78]. During the last years, the research focused its attention on understanding underlying processes of ocular neoangiogenesis for identifying angiogenesis regulators as novel therapeutic agents. In vitro models can be assessed to perform experiments modeling pathogenic angiogenesis progression in DR and AMD. These models can be used to test many drugs with antiangiogenic potentials, such as VEGF blockers [79,80]. Currently, the most common models modeling angiogenic pathologies are represented by co-culture of retinal pigmented epithelial cells, endothelial cells, or pericytes as the one developed by Eyre et al. [81]. In particular, it is composed of microvascular endothelial cells and human retinal pericyte cells cultured on either side of transwell inserts until 21 days in normoxic and hypoxic oxygen conditions and with the normal (5.5 mM) or very high glucose concentration (33 mM) to obtain control or determine a diabetic phenotype in culture [81]. To confirm the efficacy of diabetic induction, the release levels of the endothelial growth factors, including VEGF, angiopoietin 2, and platelet-derived growth factor were analyzed, and the induction of pathological conditions was highlighted by higher levels of these factors compared with control conditions. At the same time, a decrease in human hepatocyte growth factor, tissue inhibitor of metalloproteinases 2, and interleukin 8 upon diabetic conditions was observed. These results confirm that this model is suitable for testing novel pharmaceutical interventions aimed at targeting the early stages of DR [81].

In the last decade, numerous angiogenesis models were characterized to mimic all of the steps of the neoangiogenic progression occurring in DR: endothelial cell proliferation, migration, extracellular matrix (ECM) invasion, and vasculogenesis to allow rapid screening of a large scale of anti-angiogenic compounds in all of the disease stages [82]. These models use microvascular and macrovascular endothelial cell lines. Notably, over the years, human umbilical endothelial cells (HUVECs) were widely used to observe the capacity of VEGF-Trap, bevacizumab, and ranibizumab in inhibiting VEGF-induced endothelial cells proliferation, migration, receptor activation, and signaling [83,84].

A tissue-engineered choroid model was characterized to obtain a model to replicate the choroidal stroma in AMD using human choroidal stromal fibroblasts assembled in an extracellular matrix. RPE cells, HUVECs, and choroidal melanocytes were placed on the top of ECM. While RPE cells develop in a 2D monolayer on top of the matrix, HUVECs were also able to form tubular structures that generate a complex vascular network that could be suitable to study the effects of anti-neovascularization agents [85].

Another model was assessed by Shokoohmand et al. [86]. Bruch’s membrane was generated with a scaffold of poly(ε-caprolactone) gelatin electrospun and laminin.

Specifically, monkey choroidal endothelial cells were first seeded on the bottom of the scaffold, and after 6 DIV, human RPE cells were seeded on the top. The cells were then co-cultured for 20 days and numerous experiments were performed. It was observed that RPE cells maintained their phagocytic functions even if the choroid layer produced more VEGF and PEDF compared to a cell monolayer. This experimental procedure mimics the shift of VEGF/PEDF production in the early stages of AMD and validates the need for 3D models as more efficient for AMD in vitro studies compared to 2D approaches [86].

### 4.2. Blood–Retinal Barrier Models

Another typical dysfunction of DR occurs at the BRB level [87]. For this reason, numerous in vitro models were validated to mimic the in vivo conditions, which represent valuable tools to study the impact of high glucose at the BRB level as well as the trafficking between barrier sides or to develop novel molecules with beneficial effects against the BRB injury. An in vitro model of human BRB uses retinal pericytes, retinal astrocytes, and endothelial cells applicating the same in vivo layer order and the numerical ratio to mimic the physiological conditions [88]. After the assessment of the barrier, it was shown that glucose exposure induced BRB breakdown, increased barrier permeability, and reduced the expression of junctional proteins [88]. In addition, an improved expression of pro-inflammatory mediators and enzymes participating in the oxidative stress response, together with an important increase in reactive oxygen species production, was observed [89]. Furthermore, researchers show activation in immune response pathways of heme oxygenase 1 and erythroid 2–related factor 2. Overall, all of the conditions presented replicated the effects of high glucose induced in the human retina [90].

Such BRB models can be used also to model the progression of wet AMD by replicating the damage induced by choroidal neovascularization, leading to edema in the retina, photoreceptor damage, and RPE detachments, due to vascular leakage and the thickness of the Bruch’s membrane [91].

Since many factors determine AMD, to more accurately replicate the conditions in vitro, some studies focused on creating cell co-cultures, or culturing cells on an artificial Bruch’s membrane. The most used substrates for RPE culture are collagen I and IV, fibronectin, Matrigel, and gelatin. Collagen creates a layered structure with oriented fibers that mimic Bruch’s membrane and increases the functionality of RPE cells [92].

A functional BRB has been assessed with RPE cells and human vascular cells separated by an amniotic membrane obtained from human female donors by cesarean section to mimic Bruch’s membrane [90]. Artificial Bruch’s membranes can also be constructed using fibroin, a silk protein, which is a good candidate for the culture support of RPE cells and microvascular endothelial cells [93,94]. Another crucial component of AMD is represented by microglial cells that participate in the inflammation migrating to the subretinal spaces. RPE cell lines can be cultured with conditioned medium from activated microglia cells inducing, in RPE cells, an accumulation of lipids, autophagy, and expression of pro-inflammatory genes, such as interleukin-6 (IL-6), interleukin-8 (IL-8),granulocyte-macrophage-colony stimulating factor (GM-CSF), interleukin-1 beta (IL-1b), CC-chemokine ligand 2 (CCL-2) [95]. Moreover, primary RPE cells co-cultured with LPS-activated microglia cells determine the release of pro-inflammatory cytokines, showing lower levels of junctional proteins, lower levels of RPE65, and the alteration of cellular morphology [96].

The combination of the described BRB models and the co-culture with microglial cells could represent an in vitro depiction of pathological events, in particular the inflammation, which occurs in AMD and DR, to investigate cell–cell communication and the effects of the newest anti-inflammatory pharmacological approaches.

### 4.3. Retinal Pigmented Epithelium Cultures

As previously described, the RPE is the retinal layer involved in many retinal diseases. In dry AMD, the formation and accumulation of drusen that block the oxygen flow from the choroid to the RPE, is observed. In the late stages of DR diabetes, hypoxia, VEGF release, and neo-vascularization determine the breakdown of the RPE barrier [97]. Moreover, in some pathological conditions, including RP, RPE cells lose junctional complexes, and proliferate and migrate reaching the vitreous cavity.

To model the diseases previously described, researchers have proposed the primary RPE cells. They can be isolated from both human adult eye cups and human fetal eyes. Cells are obtained after the dissociation of fetal eye cups (10–22 weeks) and are variable in different donors [98]. Postnatal RPE cells have a more limited expansion potential compared to fetal RPE cells [99]. The main disadvantage of primary cultures is the limited capability to proliferate under culture conditions and usually acquire mesenchymal phenotype over time. For these reasons, in the past, researchers focused their attention on recreating alternative in vitro models of the RPE using the human immortalized retinal pigment epithelial cell lines, such as ARPE-19 (established by Dunn et al. [100]) and hTERT-RPE1 [101,102], a telomerase-immortalized RPE cell line. ARPE-19 cells present some disadvantages represented by slow differentiation properties not comparable between different laboratories; moreover, it has been shown that the original phenotype is modified among passaging and it has been proven that the ARPE-19 gene expression is different among several culture conditions [103]. Several reports indicate hTERT-RPE1 as a more suitable cell line for RPE modeling. The hTERT-immortalized retinal pigment epithelial cell line, hTERT RPE-1, was derived by transfecting the RPE-340 cell line with the pGRN145 hTERT-expressing plasmid. The cells express the Ep-16 antigen and cytokeratins. These cells have been proven to be suitable models for RPE attachment on Bruch’s membrane as well as a model to study oxidative stress response [104]. However, in a previous study, the proteome of ARPE-19 and hTERT were analyzed, comparing their constitutive and *de novo* synthesized protein expression profiles to human early passage retinal pigment epithelial cells (epRPE) by 2D electrophoresis and MALDI-TOF. The comparison between cells revealed higher levels of proteins related to cell migration, adhesion, and extracellular matrix formation, paralleled by low levels of proteins involved in cell polarization, and showed an altered expression of detoxification enzymes in hTERT-RPE. ARPE-19 cells showed a higher expression of components of the microtubule cytoskeleton and differences in the expression of proteins related to proliferation and cell death. epRPE cells, hTERT-RPE, and ARPE-19 and, therefore, may respond differently to certain functional properties, depending on the parameters to be studied [105].

A more recent option is to differentiate RPE cells from embryonic stem cells (ESCs). This differentiation was first shown for primate ESCs [106]. Cells were cultured on PA6 feeders, and the differentiation was observed after 3 weeks. Differentiated cells showed a polygonal morphology and were positive for RPE markers, such as ZO-1, RPE65, and cellular retinaldehyde-binding protein [107]. Further studies were performed in hESC cultures on mouse embryonic fibroblast feeders, and the mRNA analysis of RPE cell lines derived from a large number of hESC lines showed that the gene expression of the ESC-derived RPE cells was more similar to adult RPE cells than the immortalized RPE cell lines. The CR-4 hESC is a cell line that was derived from the inner cell mass of a developing blastocyst. This is a young and robust cell line with ample differentiation capabilities and very high expression of pluripotency markers. These cells were used to derive retinal pigment epithelium (RPE) with the aim of modeling AMD [108]. After a robust differentiation, this cell line acquires cell surface markers to generate a functionally competent RPE. Moreover, these cells were also capable of phagocytosis, a crucial RPE function. More recently, similar results were obtained from iPS-derived RPE cells [109,110]. Overall, we can conclude that, among the available models for RPE modeling, in vitro models constituted by primary RPE cells isolated from human adult eye cups and human fetal eyes, as well as RPE cells from ESCs, demonstrate high similarity with native RPE, and are more suitable for drug screening and transepithelial assays.

### 4.4. In Vitro Models of AMD

AMD is caused by the interaction of genetic and environmental factors that compromise the RPE function, leading to photoreceptor cell death and alterations in the central vision. Cellular models are useful tools in recapitulating the AMD pathological aspects due to the unique features of the human eye, where in vivo models fall into mimic AMD pathophysiology [111].

Nowadays, the availability of stem-cell technologies show huge potential for replicating the aspects of AMD [112].

Physiologically, the apical RPE surface contacts retinal photoreceptors while the basal region is attached with the Bruch’s membrane [9]. Then, RPE cells in culture need analogous substrates to differentiate and acquire epithelial functions [113]. The ECM of the RPE is critical for in vitro models and therapies. ECM composition and purified proteins, such as vitronectin, laminin, and collagen IV impact the RPE growth, pigmentation, and barrier functions [114]. In particular, purified ECM proteins can differentiate induced pluripotent stem cells (iPSCs) in the RPE phenotype [115]. Another tool is represented by a bioengineered polymer that supports matrices to ameliorate stem-cell viability and differentiation [116]. Moreover, support matrices induce the establishment of a single layer of polarized RPE cells with specified basal and apical sides [117,118]. The characterization of a polarized RPE in vitro permits drug testing to analyze molecular transport and secretion [119]. The formation of sub-RPE deposits is a common feature of several pathological conditions, in particular in AMD. Cellular models enable replication of the formation of those deposits [120]. RPE cells, when grown on porous supports, produce subcellular deposits, when treated with human serum, comprising drusen-associated molecules and activated complement proteins. Notably, this model can be used to test new drugs to inhibit the activation of the complement system and researchers demonstrated that when RPE cells are defective for the Factor-H, a crucial regulator of the complement system, are more sensitive to the complement system activation when treated with toxic metabolites from photoreceptor outer segments [121].

Then, human autologous iPSCs can be isolated to obtain RPE cultures for in vitro models to test personalized treatments as well as obtain a more accurate model of the patients’ disease conditions from AMD patients [122,123]. Notably, using an iPSC-RPE model, scientists found that AMD-associated abnormal expression at the ARMS2/HTRA1 locus disturbed cells’ antioxidants mechanisms, in particular, low SOD2 activity against oxidative damage in the RPE. SOD2 defense is impaired in RPE homozygous for the risk haplotype, while there are fewer consequences in RPE homozygous for the protective haplotype [124].

The main weakness of this kind of model is the fact that AMD involves several systemic dysfunctions that could not be represented. For this reason, to better reproduce physiological conditions and investigate the role of the interaction between cells, more complex in vitro structures were developed. Thus, RPE cultures can be combined with different retinal cells or an in vitro representation of the choroid capillary bed [125].

### 4.5. In Vitro Models of Glaucoma

The main feature of glaucoma is the establishment of an elevated intraocular pressure; thus, in vitro models of elevated hydrostatic pressure (EHP) have been used as a model for this disorder.

Numerous models evaluate the impact of elevated pressure on cells with several time points and pressure values. Clinically, the values of intraocular pressure (IOP) ranging from 20 to 35 mm Hg [126,127].

To investigate the mechanisms leading to glaucomatous RGC death, in vitro models must replicate the consequences of elevated pressure for limited periods. Many works report IOP injury with pressures between 30 and 100 mm Hg above atmospheric pressure, in a period ranging from 10 min to 72 h. The systems used are constituted by pressurized chambers made of thermoplastic materials (mainly poly (methyl methacrylate)) inflated with a flux of air/CO_2_ connected to a pressure controller and maintained in a temperature-controlled system. Chambers keep a continuous hydrostatic pressure ranging between −1 and +1 mm Hg to maintain the stabilization of the pressure influx into the chamber, in parallel, control cells are maintained at atmospheric pressure values (760 mm Hg) in a normal environment, or sometimes in normotensive conditions (15 mm Hg). Different kinds of cell lines were used to perform EHP experiments: B35, PC12, Müller cell cultures, microglial cell cultures, or human optic nerve head (ONH) astrocytes [128,129,130].

Several studies using the pressure chambers demonstrated that the injury of neuronal cells B35 and PC12 to 100 mm Hg for 2 h induces apoptotic death [100]. Moreover, the exposure of PC12 cells to pressure values ranging from 15 to 70 mm Hg for 24 h triggers apoptotic death, oxidative stress, mitochondrial alterations, and a decrease in adenosine triphosphate levels explaining the causal link between mitochondrial oxidative impairment occurring in early stages of glaucoma [131]. Additionally, the retinal ganglion cells’ exposure to increasing pressure values leads to a graded increase in apoptosis that is similar in the patients that present acute glaucoma (100 mm Hg) and chronic glaucoma (30 mm Hg) [132].

EHP can be induced also through cell immersion inside custom-built columns full of culture medium. Controls are cultured with the same quantity of culture medium but using horizontal columns. Using this setup, type 1B astrocytes were exposed to 15 mm Hg for 1–5 days stimulating cell proliferation and migration in combination with the production of soluble elastin fibers that have been observed in tissue remodeling occurring in glaucoma [133]. In the eye affected by glaucoma, high IOP leads to stretching, compression, and cribriform plate rearrangement in the human optic nerve head.

Lamina cribrosa cells are incessantly exposed to mechanical stress. For this reason, models inducing mechanical strain have been developed for the investigations of cell response [134]. In fact, commercially available straining systems exist that can be used to perform a programmable bi-axial strain across laminin-coated culture wells when positioned on a vacuum base station, whereas control cells are maintained in static conditions. For example, isolated human cells of lamina cribrosa, when kept under the mechanical strain of 15% stretch at 1 Hz for 24 h express numerous genes involved in cell proliferation, growth factor release, and signal transduction [135]. Moreover, mechanical changes affect the expression of genes that are involved in the remodeling of the extracellular matrix in primary cultures of human lamina cribrosa cells [136], suggesting that the described mechanical strain approaches could be a more suitable platform to further clarify the events leading to RGCs degeneration and subsequent vision loss in glaucoma.

### 4.6. 2D Models of Retinoblastoma

Numerous retinoblastoma cell lines are commercially available: RBL-30, RBL-13, RBL-383, Y-79, WERI-RB1, and RBL-15. Moreover, 2D in vitro models made of adherent cell culture are widely used to screen a wide range of anticancer molecules, but rarely replicate the clinical conditions [137]. For this reason, in the next paragraph, 3D models of retinoblastoma that comprise surface-engineered scaffolds and organoids will be discussed.

## 5. Alternative 3D In Vitro Models

### 5.1. 3D Models for the Posterior Segment of the Eye and RP

The posterior segment of the eye is represented by the back two-thirds of the eye, including the vitreous humor, the retina, the choroid, and the optic nerve. Treatments for the posterior segment of the eye diseases, including glaucoma, AMD, and DR represent a challenge for clinicians because of the complexity in eye static and dynamic barriers. The standard procedure for treatments of these diseases is the intravitreal administration of drugs [138]. Direct intravitreal injection has the advantage of readily introduce drug therapeutic concentrations into the vitreous humor avoiding systemic exposure. However, targets are usually the retina and the choroid, so the drug has to cross some barriers and the formulation could not exert its pharmacological effect inside the vitreous humor. Another disadvantage is the invasiveness of this procedure [139]. The injection may determine the patient’s discomfort, eye pain, endophthalmitis, vitreous detachment, retinal hemorrhage, and inflammation, particularly when multiple injections are necessary. Thus, there is a high need to focus on detecting novel formulations for eye diseases that maintain therapeutic activity and concentration of the drug in the vitreous, as well as decrease the number of the injection [140,141].

Therefore, it is important to recreate in vitro physiological and biochemical barriers of the posterior segment of the eye to ameliorate the drug design and accelerate the studies, avoiding the use of animal models in preclinical investigations. These kinds of models can be used to perform permeability studies and investigate metabolism, transport, or toxicology investigations, as well as mimic the disease causative events upon the treatment with stressors. The majority of vitreoretinal models are constituted by primary endothelial or epithelial cells or ex vivo isolated RPE/choroid tissues [142]. Retinal models based on primary cell cultures or explants are more suitable to recreate eye barriers; however, they are laborious and require complex isolating protocols. Employing cell lines, on the other hand, is more straightforward in culture conditions and storage, they have clearer genetics and facilitates the comparison of the results among laboratories as well as prevent the use of animals [142]. The most common and used in vitro system includes the culture of cell types participating in the structure of the BRB on specialized filters (in particular Transwells). These filters allow the coating with components mimicking extracellular matrices, such as laminin and fibronectin alone and in combination. This kind of model allows cells to set in a polarized trend and permits permeability studies across both the outer and inner BRB by quantifying the number of molecules added on the apical side that crossed the BRB and moved to the bottom side at set time points. Thus, the evaluation of the molecules that passed across the barrier can be performed with several methods ranging from fluorescence with enzyme-linked immunosorbent assay readers to mass spectrometry. In the last years, researchers have made progresses in the assessment of innovative co-cultures of different BRB cell types. In addition, compounds permeability can be determined to evaluate the potential barrier properties of the 3D model.

Furthermore, a research group stimulated human iPSCs to mimic the most important steps of retinal development, to obtain a 3D retinal tissue by manipulating intracellular signals, to induce the differentiation of the cells. Together with the native events, the human iPSC-derived aggregates, after 8 days of differentiation in a specific neural culture medium and attached on Matrigel-coated culture dish retinal progenitor cells, start expressing LHX2, an important transcription factor of the eye field. In the central region of the plate, other cells appeared that acquired an anterior neuroepithelial differentiation expressing PAX6 and SOX1. Moreover, differentiated photoreceptors start to develop within the cell culture, and outer segment discs acquire the photosensitivity function, providing a powerful model for functional, developmental, and translational studies [143]. Then, using these kinds of approaches, a structure similar to the embryonic optic cup can be obtained. This structure contains RPE and neurosensory layers that can self-organize in human stem cell-derived cultures. The reprogramming of iPSCs from a RP patient with the mutation in USH2A was performed to generate multilayered retinal organoids that included both retinal neurons and RPE [144]. The mutated 3D culture showed substantial defects in the cell morphology, immunofluorescence staining, and mRNA profiling, suggesting that this model significantly recapitulates the USH2A-associated pathogenesis in RD. RP2 mutations underlie a severe form of X-linked RP and animal models do not adequately reflect this phenotype. RP2 can be knocked out in iPSCs or iPSCs can be obtained from a patient with the same defects in the RP2 gene to produce retinal 3D organoids to model degeneration of photoreceptors occurring in RP [145].

Different studies used ESCs to obtain self-organized optic cups and storable stratified neural retinas. A research group showed that mouse ESCs using a floating culture in serum-free and growth-factor-reduced medium spontaneously generate a vesicle of continuous neuroepithelium, including the retina-forming field. Moreover, it has been assessed that mouse ESCs can form a structure mimicking the embryonic optic cup with the outer layer developing in RPE and the inner layer differentiating in the neural retina. Recently, another model was generated using human ESCs to obtain self-organized optic cups and storable stratified neural retinas.

The epithelium obtained from human ESCs can grow for a long time in culture to develop in a stratified organoid on 126 DIV. Moreover, if these differentiation processes can be controlled without resolving in a tumorigenic phenotype, such 3D tissues might be used as transplants to substitute entire sections of the injured retina [146].

Recently, numerous efforts have focused on the application of 3D inkjet bioprinting as a method to reconstruct the 3D structure of the retina [147]. One example is from a research group that showed a bioprinted retinal model, where the Bruch’s membrane was replaced with a gelatin derivative material (GelMa) that contained several matrix metalloproteinase peptide motifs. Then, RPE cells were bioprinted, taking advantage of the high resolution of the machine X/Y axis (5 μm) and the really low deposition volume (<180 pl) [148].

The continuous development of these models suggests that the ability of RPE cells to form tight junctions, to perform polarized cytokine secretion and phagocytosis, are related to the 3D conformation of the retina. This is why the future research for the regeneration of the damaged retina will be represented by tools where the conformation of 3D retinal layers and the retinal cell-secreted matrix is maintained.

### 5.2. 3D Models for Retinoblastoma

RB is a retinal tumor that usually occurs during childhood and, if not treated, can become fatal in 1–2 years. Treatments for advanced retinoblastoma target all the tumoral sites, including the retinal tumor, “vitreous seeds” and related “subretinal seeds” (associated vitreous tumors) [149,150]. Clinical studies showed that intravenously administered chemotherapeutics fail in the control of the seeds that mostly present an infiltrative and malignant phenotype. In addition, other causes of delivered treatments minor penetration are the avascular sites in the vitreous and subretinal space [151,152].

Advanced strategies were studied to obtain novel in vitro models that replicate the in vivo conditions: an example is the development of surface-engineered, permeable, and biodegradable, Poly(d,l)-lactide-co-glycolide (PLGA) microparticles employed as a scaffold for a 3D model using the Y-79 RB cell line. The research group compared the effect of chemotherapeutics between 2D and 3D conditions, and studied the influence of the scaffold in the ECM production and mi-RNA expression profile implicated in RB oncogenesis [153].

The discovery of organoid cultures determines the creation of 3D models that can self-organize and replicate the tissue architecture. Notably, it was demonstrated that tumor 3D cultures could be obtained from isolated retina, preserving molecular and cellular characteristics of the original retinoblastoma. Organoids can assume the histological structure of both retinal tumors and seeds [154].

A 3D model of RB vitreous seed was developed to investigate the chemotherapy agents’ penetration through live-cell imaging approaches. Enucleated eye cultures from RB patients were established and correlated with the human pathological tissues. Confocal microscopy was performed to follow the topotecan route inside several sphere sizes, and, in parallel, cell viability was evaluated. The results displayed that the obtained spheres were able to mimic the histology, phenotype, and genotype observed before in patients and an indirect correlation between sphere size and the time for the drug penetration inside the core [155]; thus, confirming that this can represent a promising model for drug distribution in future investigations.

In another study, isolated samples from not-treated patients were dissociated and cells were mixed with Matrigel solution and cultured with a specific culture media able to support the growth of neural progenitors (rich in epidermal growth factor (EGF) and fibroblast growth factor (FGF) supplements). The applied method induces the formation of RB organoids that can be expanded in vitro for over eight passages. Moreover, in this case, researchers evaluated the organoid’s drug sensitivities derived from different RB stages and found that organoids replicate the histology of retinal tumors and related seeds. Moreover, RB organoids maintain alterations in DNA copy number and express the same genes and proteins of the original tissue. The main cell populations are represented by M/L positive cells (participating in the cone signal circuitry) and glial cells that express the glial fibrillary acidic protein (GFAP). The responses from drug treatments were comparable with those observed in patients with advanced forms of RB. Therefore, this study supports patient-derived organoids as a more realistic model compared to 2D cultures to screen a wide range of drugs against advanced RB progression [156].

## 6. Conclusions

In recent years, great efforts were made to generate suitable in vitro models to (preliminarily) test potential treatments and minimize the use of animals. Besides the numerous ethical and regulatory issues, in vivo studies are expensive and time-consuming, and few animal models recognize the high translational value. Indeed, even if rodents were largely used as retinal disease models, they do not represent a suitable choice because of the differences with the human eye anatomy [157].

In vitro analyses are straightforward research approaches. Investigators may discover analyses that are more detailed by studying the biological effects and possible toxicity in a larger number of conditions before translating in humans. Experiments with cellular models present numerous benefits are cheaper, easier to use, more modulable and faster. In particular, the broad-scale evaluation of a wide range of molecules may be achieved by applying the in vitro studies to drug discovery [158].

As previously mentioned, there are numerous models used to study eye disorders. The main weakness of conventional 2D in vitro models involves their low efficacy in reproducing the complexity of dialogues amongst different cell classes and matrices. Thanks to emerging 3D cell/tissue models, these concerns have been addressed.

For this reason, the present review focuses its attention on the emerging 3D stem-cell-based in vitro protocols as more promising for disease modeling and drug discovery [159]. In particular, the evidence collected, to date, from the 3D cultures that exploit ESCs and iPSCs was presented, by demonstrating their fidelity in reproducing the main aspects of retinal complexity under physiological or pathological conditions.

Regarding the organoids, despite the remarkable research outcomes, many aspects need to be addressed to standardize the in vitro retinal model, and to make it closer to clinical conditions, thus overcoming most of the weaknesses. In this context, for example, the long culture time and the unformed photoreceptor outer segment represent, to date, a limit [160]. In the near future, organoid reproducibility needs to be improved by introducing immune cells and vascularization. To standardize the systems and make their manipulation easier, retinal 3D organoids should be included in an organ (on chip technology) [161,162].

The development of 3D in vitro models that more closely replicate in vivo conditions and support 2D and animal studies will encourage scientists to become more confident that the translation in humans will be successful.

## Figures and Tables

**Figure 1 ijms-22-10334-f001:**
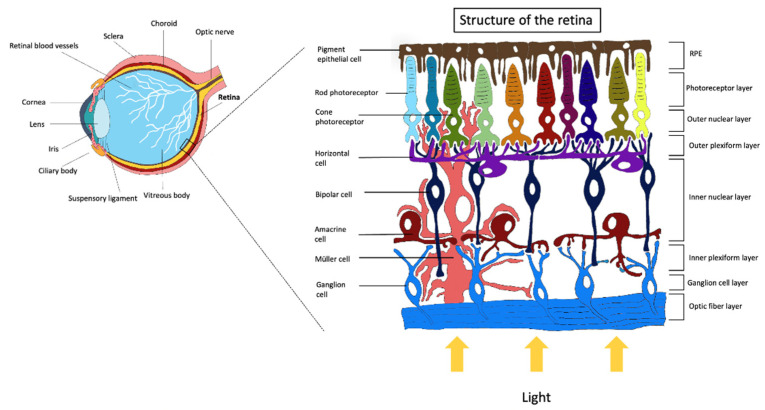
Localization of the retina in the human eye and a representation of retinal layers and associated cells.

## Data Availability

Not applicable.

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
