# Peer review of "Looking for In Vitro Models for Retinal Diseases"

_ijms, 2021, doi:10.3390/ijms221910334_

Round 1

Reviewer 1 Report

Overall impression:

Alfonsetti et al. have investigated in vitro models for the study of retinal pathologies and the development of novel therapies. They propose several alternative two-dimensional (2D) and three-dimensional (3D) in vitro models that can replicate the different cellular and matrix components of retinal layers and the injuries that lead to pathologies and loss of the visual function. 

The review tries to give an overview on available in vitro models for retinal disease. While this is a relevant topic to the field, there are several aspects that need to be addressed:

  1. From a lingual point of view, the manuscript does not meet the standards needed for publication. Frequent mistakes, poor grammar, can inappropriate choice of expressions and partially incomprehensible sentences bring doubt whether the manuscript was properly proof-read before submission.
  2. The introduction fails to provide a smooth transition into to the topic.
  3. The chosen diseases are just an excerpt of relevant pathologies and seem to be a bit random in selection.
  4. The display of in vitro models is not exhaustive and several important models are missing. Further, some of the mentioned models are not state-of-the-art (e.g. ARPE-19 vs. hTERT-RPE1)
  5. The advantages and disadvantages of 2/3-D models were not described in detail. The pathologies and causes of the diseases were not mentioned in detail and clearly. The animal models for the retinal pathologies were not described properly. Some references are missing.

Taken together, the manuscript in this form is not suitable for publication and major changes are obligatorily needed.

the abstract may be revised to underline why we really need 2D or 3D alternatives to study retinal disease.

Introduction

Some sentences contain certain points, but the reason was not explained:

For example: line 52- “the connection between RPE and photoreceptors activity” was not explained.

Line 54- retinal pathologies (which ones?), line-76- rewrite the sentence (in the human retina exist…). Line 87- Retinal dysfunctions… death of one more of the cells… (which cells?).

  • Retinal cells and function

In total too low level for a review. Also, it does not well in giving an introduction to the topic.

Line 30: “Retina is the deepest, light-sensitive region of the eye”

Deep is not a defined term in anatomy. However deep would be defined, the retina won´t be the “deepest” region of the eye. Further, the retina as a whole is not light-sensitive, only photoreceptors are.

Line 40: “Choroid is the first retinal layer, 43 located inside the eye sclera…”

The choroid is not part of the retina.

Line 50: “outer structures The choroid”

Dot missing

Line 77 ff.: “Rods are characterized by a major sensitivity to light compared to cones and thus it allows the vision of the eye under low-light levels. Cones are differentiated by the sensitivity to different light wavelengths and compared with rods, show a faster response during phototransduction, and work better in conditions of bright light…”

Not fundamentally wrong but not a proper description.

Figure 1: Localization of the retina in the human eye and a representation of retinal layers and associated cells.

According to the picture, the retina is surrounding the optic nerve, which it is not. The choroid is too thin compared to the retina. Pigmented layer should be called RPE. Display of the conjunctiva is misleading. What do the white lines inside the eye represent?

Line 94: “…Therefore, retina is a tissue that metabolically consumes high levels of nutrients and oxygen…”

Therefore does not work here, because no previous arguments were made backing this.

Section 2: A brief introduction about retinal diseases and the most common retinal vascular diseases, etc. should be given before the subtitles.

Section 2.1 the authors have reported DR and AMD (wet) in this section. In vivo models are not well-explained for AMD. Dry AMD was not described. Only one model was described for DR, although several exist (ref 28-Olivares et al). Gene therapies for AMD were mentioned but not explained. Animal models for AMD were not described.

Section 2.2 line 136-137-  What rodent models are available for glaucoma? For Retinitis pigmentosa, the P23H model, the most common mutation in the US and animal model for studying this disease, was not mentioned. RP models such as mice, rodents, etc.. were mentioned, but it was not explained which ones and what they could be used for.

Section 2.3 Retinal tumors…Please explain the animal models for retinal tumors in detail.

 Line 106: “In vivo models of DR include animals pancreatectomized or treated with alloxan [26] or streptozotocin (STZ) to damage pancreatic β-cells that produce insulin and animals ex-107 posed to high-galactose diets and eyes laser-induced or chemical-mediated injury”

Check sentence structure.

Line 109: “ ..because it is characterized by a fastest diabetic progression compared..”

THE fastest…

Line 111: “Another pathology that involves not only the retina but also the interconnected tis-111 sues such as the choroid and the RPE is the age-related macular degeneration (AMD). This 112 is a common disease of the old in the western world involving both the retina and near 113 tissues such as the choroid and the RPE”

Repetitive.

Line 115: “The pathogenesis starts with the deposition of waste material at the RPE level inducing the non-neovascular form 116 that can develop into an exudative form that is called “wet” leading to a neovascular disease [30]”

Clinically this is not an accurate description of AMD.

Line 126: “ While glaucoma involves mostly the ganglion cell layer”

Glaucoma predominantly damages the RNFL, not GCL. If you refer to the ganglion cells and not the GCL the statement would be correct, although it would be better to refer to the RNFL. 

Besides, glaucoma is usually not classified as a retinal degeneration. Having RP and glaucoma in the same paragraph does not make sense.

Line 162: “This tumor is most of the time bilat-162 erally and it is frequently multifocal. Retinoblastoma”

No specific tumor was mentioned before.

Table 1: Most common retinal disorders and relative clinical features that can be represented by in vitro models.

HUVECs migration assay (Boyden chamber). It is not an in vitro model, this is an in vitro assay. Please change the table according to the in vitro cultures and assays.

General: This table in this form is considered to be of limited use.

Section 3. From animal models to in vitro studies

Please improve the sentence. (line 182) “ investigations for animal models of human retinal disorders determine many recognizable benefits (what kind of benefits??). Molecular studies permit to find new potential candidate genes (which genes??) that play a role in human retinal diseases that were unrecognized in the past...“.

Section 4: Alternative 2D in vitro models

4.1 Please describe the 2D model and the diseases for the neoangiogenesis.

Line 222: A culture model developed by Eyre et al. Which model is it? Please put the citation?

As they wrote in line 219 in vitro models for DR and AMD for neuroangiogenesis, I could not see any in vitro models for AMD. Moreover, DR animal models were not clearly explained.

4.2 Blood retinal barrier models: Line 250-251: “induced BRB breakdown, an increase in barrier permeability together 250 with a reduction in the expression of junction proteins...“ Reference is missing.

Line 257- “ Such BRB models can be used also to model the progression of wet AMD by replicating the pathological conditions between RPE cells, Bruch's membrane, and vascular cells..“. Please explain how and why can this model mimic AMD? Please clearly mention about the pathologies of AMD, thickness of the Bruch’s membrane, drusen accumulation and then the in vitro models. Therefore, readers can get the connections between the pathologies of the diseases and the interested in vitro models. 

4.3 Retinal pigmented epithelium cultures:

Please rewrite the sentences below;

Line 264- Another important alteration occurring not only in DR but also in AMD and RP is at the level of the RPE.

Line 266- “However, the isolation needs specific expertise and the use of several animal resources and at the same time is more difficult to compare data between different research groups, where the use of a cell line facilitates this comparison [74].”

Line 270- “For this reason, researchers focused their attention on recreating in vitro models of the RPE using the human retinal pigment epithelial cell line ARPE-19 established by Dunn et al.” ARPE-19 cells are not accepted as a good RPE model (differentiation, the original phenotype of the cell line has undergone a degradation among passaging, loss of RPE specific phenotype and variabilities in different culture conditions (Hellinen et al., 2019). Please put the citation. Please mention about human immortalized RPE1 cells (hTERT-RPE1), which are also not perfect but better model for investigating AMD.

4.4. In vitro models of AMD:

Please rewrite the sentences below,

Line 288-“The AMD is a complex disease affecting several regions of the eye and because of the distinctive characteristics of the human eye animal models are not the best models to replicate the pathology [80]”.

Line 318- “scientists found that AMD-associated gene variants (ARMS2 and ARMS2) (?) disturbed cells’ antioxidants mechanisms…”. What are the other gene variants or polymorphisms? Which antioxidant mechanisms?

4.5 In vitro models of glaucoma:

Please rewrite the sentences; Line 369- “To work with these models, exist commercially available straining systems that can be used to perform constant tension or adaptable stimuli whereas control cells are maintained in static conditions.

Line 373 “Moreover, mechanical changes affect the expression of genes that are involved in the remodeling of the extracellular matrix that appears blocked by the obstruction of calcium channels in glaucomatous conditions [108].”

  1. Alternative 3D in vitro models:

Please rewrite the sentences; Line 379- “The most frequent route of administration for drugs targeting retinal pathologies is the posterior segment of the eye, despite the invasiveness and the limitations in drugs half-life and poor tissue permeation, throughout intravitreal or subretinal injections [109–111].” What are these invasions and limitations? Please first describe the posterior segment of the eye and posterior segment eye diseases.

Line 442- “Another similar study uses embryonic stem cells (ESCs) to obtain self-organized optic cups and storable stratified neural retinas that contain both rods and cones. Photoreceptors’ differentiation can be accelerated through Notch inhibition [119]. “Please stress the usage of embryonic stem cells.

Line 455- “Numerous retinoblastoma cell lines are commercially available: RBL-30, RBL-13, RBL-383, Y-79, WERI-Rb1, and RBL-15. 2D in vitro models made of adherent cell culture are widely used to screen a wide range of anticancer molecules but rarely replicate the clinical conditions [124].” Please diminish the 2D and 3D models and place them under the appropriate subtitles.

Author Response

Reviewer 1

Overall impression:

Alfonsetti et al. have investigated in vitro models for the study of retinal pathologies and the development of novel therapies. They propose several alternative two-dimensional (2D) and three-dimensional (3D) in vitro models that can replicate the different cellular and matrix components of retinal layers and the injuries that lead to pathologies and loss of the visual function. 

Response: We would like to thank Reviewer 1 for the comments and the time spent in reading our manuscript and for the numerous comments provided that helped in improving our research article. We tried to address all the points raised.

The review tries to give an overview on available in vitro models for retinal disease. While this is a relevant topic to the field, there are several aspects that need to be addressed:

  1. From a lingual point of view, the manuscript does not meet the standards needed for publication. Frequent mistakes, poor grammar, can inappropriate choice of expressions and partially incomprehensible sentences bring doubt whether the manuscript was properly proof-read before submission.

Response: Thank you for the comment. We now changed some parts of the manuscript to make them more readable and revised English grammar mistakes and incomprehensible sentences as suggested. The ms was revised by Grammarly software and by a naïve English speaker.

  1. The introduction fails to provide a smooth transition into to the topic.

             Response: Thank you for the comment. We have modified the introduction, hoping to address the reviewer’s suggestion.

  1. The chosen diseases are just an excerpt of relevant pathologies and seem to be a bit random in selection.

Response: Thank you for the comment. The choice was made considering the most frequent retinal diseases and the currently available in vitro and in vivo models, also in consideration of the available literature. We have now implemented their description, as suggested.

  1. The display of in vitro models is not exhaustive and several important models are missing. Further, some of the mentioned models are not state-of-the-art (e.g. ARPE-19 vs. hTERT-RPE1)

Response: Thank you for the comment. We now added critical points about the use of ARPE-19, which are now treated only from a historical point of view, and introduced hTERT-RPE1 and other cellular models as suggested.

  1. The advantages and disadvantages of 2/3-D models were not described in detail. The pathologies and causes of the diseases were not mentioned in detail and clearly. The animal models for the retinal pathologies were not described properly. Some references are missing.

Response: Thank you for the comment. We now added more information about the advantages and disadvantages of 2/3-D models, animal models for retinal pathologies, and more references were introduced in the text.

Taken together, the manuscript in this form is not suitable for publication and major changes are obligatorily needed.

the abstract may be revised to underline why we really need 2D or 3D alternatives to study retinal disease.

 Response: Thank you for the comment. We now modified the abstract according to the suggestion.

Introduction

Some sentences contain certain points, but the reason was not explained:

For example: line 52- “the connection between RPE and photoreceptors activity” was not explained.

Line 54- retinal pathologies (which ones?), line-76- rewrite the sentence (in the human retina exist…). Line 87- Retinal dysfunctions… death of one more of the cells… (which cells?).

 Response: Thank you for the comment. We now provided the information requested and modified the sentence.

  • Retinal cells and function

In total too low level for a review. Also, it does not well in giving an introduction to the topic.

Line 30: “Retina is the deepest, light-sensitive region of the eye”

Deep is not a defined term in anatomy. However deep would be defined, the retina won´t be the “deepest” region of the eye. Further, the retina as a whole is not light-sensitive, only photoreceptors are.

Line 40: “Choroid is the first retinal layer, 43 located inside the eye sclera…”

The choroid is not part of the retina.

Response: We appreciate the reviewer’s comment and modified the improper information into the paragraph accordingly to the comments.

Line 50: “outer structures The choroid”

Dot missing

Response: We apologize for the oversight. We now added the dot.

Line 77 ff.: “Rods are characterized by a major sensitivity to light compared to cones and thus it allows the vision of the eye under low-light levels. Cones are differentiated by the sensitivity to different light wavelengths and compared with rods, show a faster response during phototransduction, and work better in conditions of bright light…”

Not fundamentally wrong but not a proper description.

Response: We appreciate the reviewer’s comment, and we now modified the rods and cones description in this section.

Figure 1: Localization of the retina in the human eye and a representation of retinal layers and associated cells.

According to the picture, the retina is surrounding the optic nerve, which it is not. The choroid is too thin compared to the retina. Pigmented layer should be called RPE. Display of the conjunctiva is misleading. What do the white lines inside the eye represent?

Response: We appreciate the reviewer’s comment, and we now modified the image accordingly. Regarding the optic nerve, the picture wants to represent the retinal ganglion cell axons that converge into the optic nerve.

Line 94: “…Therefore, retina is a tissue that metabolically consumes high levels of nutrients and oxygen…”

Therefore does not work here, because no previous arguments were made backing this.

Response: We apologize for the oversight. We now eliminated this sentence. 

Section 2: A brief introduction about retinal diseases and the most common retinal vascular diseases, etc. should be given before the subtitles.

Response: We appreciate the reviewer’s comment, and we now added this section before the subtitles.

Section 2.1 the authors have reported DR and AMD (wet) in this section. In vivo models are not well-explained for AMD. Dry AMD was not described. Only one model was described for DR, although several exist (ref 28-Olivares et al). Gene therapies for AMD were mentioned but not explained. Animal models for AMD were not described.

Response: Thank you for the comment. We now added missing information.

Section 2.2 line 136-137-  What rodent models are available for glaucoma? For Retinitis pigmentosa, the P23H model, the most common mutation in the US and animal model for studying this disease, was not mentioned. RP models such as mice, rodents, etc.. were mentioned, but it was not explained which ones and what they could be used for.

Response: We appreciate the reviewer’s comment. We now implemented glaucoma and RP sections accordingly.

Section 2.3 Retinal tumors…Please explain the animal models for retinal tumors in detail.

Response: We appreciate the reviewer’s comment. We now focused this paragraph only on Retinoblastoma and changed the title.

 Line 106: “In vivo models of DR include animals pancreatectomized or treated with alloxan [26] or streptozotocin (STZ) to damage pancreatic β-cells that produce insulin and animals ex-107 posed to high-galactose diets and eyes laser-induced or chemical-mediated injury”

Check sentence structure.

Response: Thank you for the comment. We now changed sentence structure and added more information.

Line 109: “ ..because it is characterized by a fastest diabetic progression compared..”

THE fastest…

Response: We apologize for the oversight. We now changed the sentence accordingly.

Line 111: “Another pathology that involves not only the retina but also the interconnected tis-111 sues such as the choroid and the RPE is the age-related macular degeneration (AMD). This 112 is a common disease of the old in the western world involving both the retina and near 113 tissues such as the choroid and the RPE”

Repetitive.

Response: Thank you for the comment and we apologize for the oversight. We now eliminated repetitive information.

Line 115: “The pathogenesis starts with the deposition of waste material at the RPE level inducing the non-neovascular form 116 that can develop into an exudative form that is called “wet” leading to a neovascular disease [30]”

Clinically this is not an accurate description of AMD.

Response: We appreciate the reviewer’s comment. We tried to provide a more accurate description of AMD pathology.

Line 126: “ While glaucoma involves mostly the ganglion cell layer”

Glaucoma predominantly damages the RNFL, not GCL. If you refer to the ganglion cells and not the GCL the statement would be correct, although it would be better to refer to the RNFL. 

Response: We apologize for the oversight. In this paragraph we were referring to the ganglion cells damage, so we changed the sentence accordingly.

Besides, glaucoma is usually not classified as a retinal degeneration. Having RP and glaucoma in the same paragraph does not make sense.

Response: We appreciate the reviewer’s comment. We now generated two distinctive paragraphs for each pathology.

Line 162: “This tumor is most of the time bilat-162 erally and it is frequently multifocal. Retinoblastoma”

No specific tumor was mentioned before.

Response: We apologize for the oversight. We now added “retinoblastoma” as the subject of this sentence.

Table 1: Most common retinal disorders and relative clinical features that can be represented by in vitro models.

HUVECs migration assay (Boyden chamber). It is not an in vitro model, this is an in vitro assay. Please change the table according to the in vitro cultures and assays.

Response: Thank you for the comment. According to the following comment, we eliminated the table.

General: This table in this form is considered to be of limited use.

Response: we eliminated the table

Section 3. From animal models to in vitro studies

Please improve the sentence. (line 182) “ investigations for animal models of human retinal disorders determine many recognizable benefits (what kind of benefits??). Molecular studies permit to find new potential candidate genes (which genes??) that play a role in human retinal diseases that were unrecognized in the past...“.

 Response: Thank you for the comment. We now briefly answered all the points raised.

Section 4: Alternative 2D in vitro models

4.1 Please describe the 2D model and the diseases for the neoangiogenesis.

 Response: Thank you for the comment. We now briefly described 2D models and the diseases for the neoangiogenesis, as suggested.

Line 222: A culture model developed by Eyre et al. Which model is it? Please put the citation?

Response: Thank you for the comment. We now better describe the model and added the citation.

As they wrote in line 219 in vitro models for DR and AMD for neuroangiogenesis, I could not see any in vitro models for AMD. Moreover, DR animal models were not clearly explained.

Response: We appreciate the reviewer’s comment. We now implemented the section for in vitro models for AMD and the in vivo models of DR.

4.2 Blood retinal barrier models: Line 250-251: “induced BRB breakdown, an increase in barrier permeability together 250 with a reduction in the expression of junction proteins...“ Reference is missing.

Response: We apologize for the oversight. We now added the reference.

Line 257- “ Such BRB models can be used also to model the progression of wet AMD by replicating the pathological conditions between RPE cells, Bruch's membrane, and vascular cells..“. Please explain how and why can this model mimic AMD? Please clearly mention about the pathologies of AMD, thickness of the Bruch’s membrane, drusen accumulation and then the in vitro models. Therefore, readers can get the connections between the pathologies of the diseases and the interested in vitro models. 

Response: Thank you for the comment. We tried to address all the points and we provided more information in this section.

4.3 Retinal pigmented epithelium cultures:

Please rewrite the sentences below;

Line 264- Another important alteration occurring not only in DR but also in AMD and RP is at the level of the RPE.

Line 266- “However, the isolation needs specific expertise and the use of several animal resources and at the same time is more difficult to compare data between different research groups, where the use of a cell line facilitates this comparison [74].”

Line 270- “For this reason, researchers focused their attention on recreating in vitro models of the RPE using the human retinal pigment epithelial cell line ARPE-19 established by Dunn et al.” ARPE-19 cells are not accepted as a good RPE model (differentiation, the original phenotype of the cell line has undergone a degradation among passaging, loss of RPE specific phenotype and variabilities in different culture conditions (Hellinen et al., 2019). Please put the citation. Please mention about human immortalized RPE1 cells (hTERT-RPE1), which are also not perfect but better model for investigating AMD.

Response: Thank you for the comment. We now rewrote the sentences and we briefly discussed the disadvantages of the use of ARPE-19 and introduced hTERT-RPE1 as an alternative RPE cell line as suggested.

4.4. In vitro models of AMD:

Please rewrite the sentences below,

Line 288-“The AMD is a complex disease affecting several regions of the eye and because of the distinctive characteristics of the human eye animal models are not the best models to replicate the pathology [80]”.

Line 318- “scientists found that AMD-associated gene variants (ARMS2 and ARMS2) (?) disturbed cells’ antioxidants mechanisms…”. What are the other gene variants or polymorphisms? Which antioxidant mechanisms?

Response: We appreciate the reviewer’s comment. We now rewrote the sentences as suggested and clarified the points raised.

4.5 In vitro models of glaucoma:

Please rewrite the sentences; Line 369- “To work with these models, exist commercially available straining systems that can be used to perform constant tension or adaptable stimuli whereas control cells are maintained in static conditions.

Line 373 “Moreover, mechanical changes affect the expression of genes that are involved in the remodeling of the extracellular matrix that appears blocked by the obstruction of calcium channels in glaucomatous conditions [108].”

Response: We appreciate the reviewer’s comment. We now rewrote the sentences in the manuscript.

  1. Alternative 3D in vitro models:

Please rewrite the sentences; Line 379- “The most frequent route of administration for drugs targeting retinal pathologies is the posterior segment of the eye, despite the invasiveness and the limitations in drugs half-life and poor tissue permeation, throughout intravitreal or subretinal injections [109–111].” What are these invasions and limitations? Please first describe the posterior segment of the eye and posterior segment eye diseases.

Response: We appreciate the reviewer’s comment. We now described first the posterior segment of the eye and the related pathologies. We now rewrote the sentence in the manuscript, and we briefly discussed the invasiveness of intravitreal drug administration and the relative limitations.

Line 442- “Another similar study uses embryonic stem cells (ESCs) to obtain self-organized optic cups and storable stratified neural retinas that contain both rods and cones. Photoreceptors’ differentiation can be accelerated through Notch inhibition [119]. “Please stress the usage of embryonic stem cells.

Response: We appreciate the reviewer’s comment. We now rewrote the sentence in the manuscript and discussed in more detail the use of ESCs for the generation of optic cups and storable stratified neural retinas.

Line 455- “Numerous retinoblastoma cell lines are commercially available: RBL-30, RBL-13, RBL-383, Y-79, WERI-Rb1, and RBL-15. 2D in vitro models made of adherent cell culture are widely used to screen a wide range of anticancer molecules but rarely replicate the clinical conditions [124].” Please diminish the 2D and 3D models and place them under the appropriate subtitles.

Response: Thank you for the comment. We now placed the sentence in the 2D model paragraph of the manuscript. We did not diminish the number of models, since they are treated only generically and not in detail.

Reviewer 2 Report

This review attempts to summarize a number of  in vitro models allowing to study retina or  retinal pathologies.

The main problem, that this review  does not critically analyze existing models and their caveats, ambitiously attempts to cover a wide range of models and diseases. Some of the discussed in vitro models are outdated. Some of the terminology is wrong. Overviews of mechanisms for  ocular diseases might be improved.

Few examples.

It is generally accepted that ARPE19 cells are far away from RPE, do not express all RPE markers, do not reach relevant TEER and more fibroblasts-like. Protocols for culturing primary RPE cells, or differentiation RPE from stem cells are well established  and became standard for use in the field for serious studies. There is no need to spend so much to discuss use of ARP19 models. In general currently studies with ARPE19 cells are not fundable  now due to availability of better / more physiological culture models to study RPE.

There are dozens of papers published with a variety of mutant stem cell derived RPE which authors left out.

Use of RGC5 line became controversial many years ago due to the suspected contamination with 661W cell line . In general  studies with use RGC5 cells are not considered credible now. Many studies performed with this line left sense of uncertainty and skepticism.

Choroid is not part of the retina, authors should rewrite this part of the introduction.

There are more then 80 genes known to cause RP and photoreceptor degeneration(not 40as authors

mention) . Authors should probably cite newer reviews on the topic.

In general, review will be better if the author would choose only one topic ( e.g., modeling RPE, retina, vasculature) and either give historic prospective on in vitro modeling or

critically characterize existing cutting edge models, what  could be learned by using them and how they could be improved . This would give better more impactful review. An attempt to overview retina, ocular diseases, criticism of mouse models and in vitro models is very ambitious and hard to accomplish with limited space  without being shallow.

Author Response

Reviewer 2

This review attempts to summarize a number of  in vitro models allowing to study retina or  retinal pathologies.

Response: We thank the Reviewer for the time spent in revising our review article and for the valuable comments provided that helped us in improving our manuscript. We tried to address all the points raised.

The main problem, that this review  does not critically analyze existing models and their caveats, ambitiously attempts to cover a wide range of models and diseases. Some of the discussed in vitro models are outdated. Some of the terminology is wrong. Overviews of mechanisms for  ocular diseases might be improved.

Response: Thank you for the comment. We now critically analyzed the models proposed in the appropriate section. We introduced more recent models, we tried to improve the terminology used in the text and we also provided more information on the mechanisms leading to ocular diseases.

Few examples.

It is generally accepted that ARPE19 cells are far away from RPE, do not express all RPE markers, do not reach relevant TEER and more fibroblasts-like. Protocols for culturing primary RPE cells, or differentiation RPE from stem cells are well established  and became standard for use in the field for serious studies. There is no need to spend so much to discuss use of ARP19 models. In general currently studies with ARPE19 cells are not fundable  now due to availability of better / more physiological culture models to study RPE.

There are dozens of papers published with a variety of mutant stem cell derived RPE which authors left out.

Response: We appreciate the reviewer’s comment. We discussed the disadvantages of the use of ARPE-19 in the manuscript, which are now present only from a historical point of view. Moreover, we mentioned alternative cell lines and we briefly discussed in vitro models of RPE obtained from embryonic stem cells and induced pluripotent stem cells.

Use of RGC5 line became controversial many years ago due to the suspected contamination with 661W cell line . In general  studies with use RGC5 cells are not considered credible now. Many studies performed with this line left sense of uncertainty and skepticism.

Response: Thank you for the comment. We now eliminated RGC5 cells as a suitable platform for in vitro models accordingly.

Choroid is not part of the retina, authors should rewrite this part of the introduction.

Response: We appreciate the reviewer’s comment. We now corrected the wrong information in the manuscript accordingly. 

There are more then 80 genes known to cause RP and photoreceptor degeneration(not 40as authors

mention) . Authors should probably cite newer reviews on the topic.

Response: Thank you for the comment. We now corrected the wrong information on the number of genes involved in RP and new references were introduced for the topic.

In general, review will be better if the author would choose only one topic ( e.g., modeling RPE, retina, vasculature) and either give historic prospective on in vitro modeling or

critically characterize existing cutting edge models, what  could be learned by using them and how they could be improved . This would give better more impactful review. An attempt to overview retina, ocular diseases, criticism of mouse models and in vitro models is very ambitious and hard to accomplish with limited space  without being shallow.

Response: We agree with the reviewer that the aim was ambitious, however, to match the criticisms of this reviewer and reviewer 1, we tried to implement the review with new information and cellular models. Moreover, for each cellular model, we tried to critically analyze advantages and caveats, hoping that the ms is now more readable.

Round 2

Reviewer 2 Report

Please let English-speaker to edit ms and consider

changing title. An attempt to counter in vitro models and in vivo experimentation is  methodologically  wrong.

It is never the goal to do experiments in vivo or in vitro. 

We always answer scientific questions  and test hypotheses

using appropriate for the task in vivo and in vitro models.  

Author Response

Reviewer 2 comments:

Please let English-speaker to edit ms and consider changing title. An attempt to counter in vitro models and in vivo experimentation is methodologically wrong. It is never the goal to do experiments in vivo or in vitro. We always answer scientific questions and test hypotheses using appropriate for the task in vivo and in vitro models.  

Response: We thank the Reviewer for the comments. We apologize for the oversights, and we now carefully checked the manuscript for the English and the main message of the review has been strengthened. We also changed the title as suggested.